# Inflammatory Indices in Patients with Myocardial Infarction Complicated by Cardiogenic Shock, and Their Interconnections with SCAI Stages and Patients’ Survival: A Retrospective Study

**DOI:** 10.3390/jcm14124283

**Published:** 2025-06-16

**Authors:** Irina Kologrivova, Maria Kercheva, Oleg Panteleev, Alexey Dmitriukov, Ivan Zenkov, Tatiana Suslova, Vyacheslav Ryabov

**Affiliations:** 1Cardiology Research Institute, Tomsk National Research Medical Center, Russian Academy of Sciences, 111A Kievskaya, Tomsk 634012, Russia; cardio@cardio-tomsk.ru (O.P.); aldmn9k@mail.ru (A.D.); tes@cardio-tomsk.ru (T.S.); rvvt@cardio-tomsk.ru (V.R.); 2Cardiology Division, Siberian State Medical University, 2 Moscovsky Trakt, Tomsk 634055, Russia; mr.iva.hy@mail.ru

**Keywords:** myocardial infarction, cardiogenic shock, inflammation, survival, prognosis, SIRI, AISI, platelet-to-lymphocyte ratio

## Abstract

**Background:** Myocardial infarction complicated by cardiogenic shock (MI-CS) remains a critical condition with high mortality rates, despite advances in treatment. Systemic inflammation plays a significant role in MI-CS progression; however, its dynamics across different stages of the Society for Cardiovascular Angiography and Interventions (SCAI) classification remain poorly understood. This study aimed to evaluate indices of systemic inflammation—neutrophil–lymphocyte ratio (NLR), platelet–lymphocyte ratio (PLR), systemic immune-inflammation index (SII), systemic inflammation response index (SIRI), and aggregate index of systemic inflammation (AISI)—in MI-CS patients, correlating them with SCAI stages and survival outcomes. **Methods:** A single-center retrospective study included 132 patients with MI-CS, categorized into SCAI stages A–E. All patients were assessed for demographic, clinical, and laboratory data, procedural and treatment characteristics, MI timing, and outcomes. Complete blood count test data were used to calculate inflammatory indices and evaluate types of immune reactions. **Results:** PLR, SII, and AISI peaked at SCAI stage C and declined significantly at stage E, suggesting suppressed inflammation in advanced shock. SIRI emerged as a key prognostic marker for stage C patients, with elevated levels associated with larger infarct size, higher heart rate, and predominant innate immune activation. Patients with SIRI ≥ 3.34 had significantly lower two-year survival (log-rank test, *p* = 0.006). **Conclusions:** Inflammation indices, particularly SIRI, provide valuable prognostic insights in MI-CS, reflecting disease severity and heterogeneity of immune response. The decline in inflammatory indices at SCAI stage E may indicate immune suppression in extreme MI-CS, underscoring the need for personalized therapeutic strategies.

## 1. Introduction

Despite the advent of early revascularization and the use of optimal medical therapy in the management of patients with acute coronary syndrome (ACS) complicated by cardiogenic shock (CS), the mortality rates in this patient population remain consistently high, exceeding 40% in some cases [1,2]. One of the most common phenotypes of CS is myocardial infarction complicated by cardiogenic shock (MI-CS), accounting for up to 80% of all CS cases [3,4,5,6].

In order to improve the management of patients with CS, the latest classification was proposed in 2019 by the American Society of Cardiovascular Angiography and Intervention (SCAI). It includes grading CS into five stages from A to E, where stage A is a condition with a risk of CS development, and stage E is “extreme” CS with circulatory collapse [7]. However, the CS heterogeneity likely extends far beyond the factors being considered in SCAI staging, suggesting that MI-CS patients may require a more tailored approach [4].

Previous studies have indicated that the systemic inflammatory response, driven by both the primary damage of the heart tissue during infarction [8] and advancing multiple organ failure [9], plays an important role in the development and progression of MI-CS. Thus, assessing the degree of the systemic inflammation in patients with MI-CS could refine the diagnostic and therapeutic strategy for MI-CS, enabling a more personalized strategy in patients’ follow-up.

A number of indices allowing highly accurate prediction of the prognosis have been proposed for patients with MI-CS [10,11]. However, many of these are based on the evaluation of serum biomarkers, limiting their applicability in routine clinical practice due to practical constraints [12].

Recently, complete blood count (CBC)-based inflammation indices have gained attention as promising prognostic tools. Their evaluation appears to be relatively inexpensive, accessible, and easy to perform, while calculation is absolutely intuitive [13,14,15]. The simplest indices include neutrophil-to-lymphocyte ratio (NLR) [16], monocyte-to-lymphocyte ratio (MLR), and platelet-to-lymphocyte ratio (PLR) [17]. The most promising for assessing the severity of the systemic inflammation are complex indices of the systemic inflammatory response. The Systemic Immune Inflammation Index (SII), the Systemic Inflammation Response Index (SIRI), and the Aggregate Index of Systemic Inflammation (AISI) [18,19] (the latter is also called the pan-immune inflammation value (PIV) in several works) represent integrated assessment of the activation of inflammation, thrombogenesis, and immune response. These have proven to be efficient prognostic biomarkers in cardiovascular disorders, cancer, and infectious disease [19,20].

Previously, data on the high prognostic significance of inflammation indices in septic shock were obtained [21,22]. Despite the fact that septic and cardiogenic shock are characterized by a number of common patterns of development, differences in the etiology of these conditions do not allow immediate extrapolation of the data on the systemic inflammatory response to MI-CS [23]. Moreover, existing clinical studies investigating hematological indices of systemic inflammation in patients with shock or myocardial infarction have not generally examined the entire arsenal of currently widely used indices (NLR, PLR, MLR, SII, SIRI, AISI), but have focused exclusively on individual indicators [21,24]. In this context, we aimed to determine the most promising index of systemic inflammation for personalized management of a cohort of patients with MI-CS.

Furthermore, there is a necessity to investigate the interrelation between indices of systemic inflammation and well-established instruments for risk stratification in patients with MI-CS, such as SCAI staging, for the successful implementation of indices in the clinical routine.

Thus, the aim of the present study was to identify indices of systemic inflammation that correlate with MI-CS staging according to SCAI classification, and to determine the most promising inflammatory indices for the evaluation of short- and mid-term prognoses in MI-CS patients.

## 2. Materials and Methods

### 2.1. Study Design and Patients Characteristics

We conducted a single-center retrospective register study, comprising analysis of 1253 medical histories of patients admitted to the Department of Emergency Cardiology, Cardiology Research Institute, Tomsk NRMC between 1 January and 31 December 2020, with diagnosis of MI. Analysis did not include patients’ personal information and was approved by the local ethics committee (Biomedical Ethics Committee of Cardiology Research Institute, Tomsk NRMC; protocol #266, 22 May 2024). All the methods and procedures were applied in accordance with the Declaration of Helsinki (2000 edition) and “Rules of Clinical Practice in the Russian Federation”, approved by the Order of the Ministry of Health of the Russian Federation in 19 June 2003 No. 266. No cases of informed consent withdrawal were recorded.

MI was classified according to the Fourth Universal Definition of Myocardial Infarction, with confirmation of the status of ST-segment elevation myocardial infarction (STEMI) or non-ST-segment elevation myocardial infarction (NSTEMI). The diagnosis and treatment of MI in all patients followed the current ESC Guidelines for the Management of Acute Coronary Syndromes [25]. Cardiogenic shock (CS) as a complication of MI was diagnosed in 132 patients based on ICD-10: a combination of arterial hypotension (SBP less than 90 mm Hg) with normal cardiac volume load with signs of organ and tissue hypoperfusion (cold extremities, impaired consciousness, dizziness, metabolic acidosis, increased serum lactate levels, increased serum creatinine levels). All CS patients were retrospectively classified according to the Society for Cardiovascular Angiography and Interventions (SCAI) staging system: stage B—“beginning” CS, stage C—“classical” CS, stage D—“deteriorating” CS, stage E—“extremis” CS. Additionally, 7 patients were assigned to Stage A, defined by the presence of acute primary anterior Q-wave STEMI with a high risk of developing CS [26]. All the patients received the standard medication for ACS in accordance with the current ESC recommendations: dual anti-platelet therapy (acetylsalicylic acid + clopidogrel/ticagrelor/prasugrel), unfractionated heparin, short-acting nitrates (nitroglycerin/nitrospray), and narcotic analgesic (morphine).

Reperfusion therapy decisions were made for patients with ACS with persistent ST segment elevation was performed immediately after ACS diagnosis. The patient was transported to the Department of Invasive Cardiology for primary percutaneous coronary intervention (PCI) of the infarct-related artery as part of the primary invasive strategy for MI treatment, provided the expected time from patient arrival to catheterization laboratory did not exceed 120 min. If the expected time of patient delivery to the Department of Invasive Cardiology exceeded 120 min, the patient underwent thrombolysis with tenecteplase as part of the pharmacoinvasive strategy at the pre-hospital stage of care, after which the patient was delivered to the cath lab for the planned or rescue PCI. The target time from hospitalization to mechanical recanalization of the infarction-related coronary did not exceed 60 min.

All patients were assessed for demographic, clinical and laboratory data, procedural and treatment characteristics, MI timing, and outcomes.

### 2.2. Evaluation of Immune Response Reactions

The type of immune reaction was determined based on the absolute counts of leukocytes and lymphocytes retrieved from the total blood count test [27]. Assignment of each particular case to a definite type of immune reaction was based on the algorithm depicted in Table 1.

### 2.3. Calculation of Indices of Systemic Inflammation

Indices of systemic inflammation were calculated based on the total blood count test results: neutrophil-to-lymphocyte ratio (NLR); platelet-to-lymphocyte ratio (PLR); monocyte-to-lymphocyte ratio (MLR); systemic immune inflammation index (SII); systemic inflammation response index (SIRI); aggregated index of systemic inflammation (AISI).

The following equations were used:NLR = NEU ÷ LYMPH;MLR = MON ÷ LYMPH;PLR = PL ÷ LYMPH;SIRI = NEU × MON ÷ LYMPH;SII = NEU × PL ÷ LYMPH;AISI = NEU × MON × PL ÷ LYMPH,where the following abbreviations apply:
NEU—absolute neutrophil count;MON—absolute monocyte count;PL—absolute platelet count;LYMPH—absolute lymphocyte count.

### 2.4. Statistical Analysis

Quantitative data are presented as median (Me) and interquartile range (Q1; Q3). Qualitative variables are presented as absolute (*n*) and relative (%) counts. The normality of the data distribution was assessed by the Shapiro–Wilk test. The Kruskal–Wallis ANOVA test was used to analyze differences between multiple independent groups, followed by pairwise comparisons of groups using the Mann–Whitney U test with Bonferroni correction. Either Pearson’s chi-square test or Fisher’s exact test was used for the evaluation of differences between categorical parameters. Correlations between variables were assessed using Spearman’s rank correlation coefficient. The Kaplan–Meier method was used to calculate the survival rate. Survival comparisons were made using the log-rank test. The prognostic significance of variables predicting the value of SIRI was evaluated using multiple logistic regression analysis. In all the tests, a *p*-value of < 0.05 was considered to be statistically significant. All the statistical calculations were conducted in STATISTICA 10.0 software (StatSoft Inc., Tulsa, OK, USA).

## 3. Results

### 3.1. Baseline Characteristics of Patients

Characteristics of patients included in the study are presented in Table 2. Patients at stages D and E were older, while the proportion of men was higher than women among patients at stages A and B. Lactate concentrations increased as expected with the progression of MI-CS stages, and pH levels decreased correspondingly. All other baseline characteristics of MI-CS patients across different SCAI stages were comparable (Table 2).

### 3.2. Complete Blood Count Test and Types of Immune Reactions

Table 3 presents data from the complete blood count test of MI-CS patients at different SCAI stages. Monocyte counts were highest in patients at SCAI stage C and significantly higher compared to MI-CS patients at stage D (*p* = 0.007). Other parameters did not differ significantly between the groups.

When analyzing the types of immune reactions, we found a high degree of heterogeneity in this parameter depending on the SCAI stage, which, however, did not reach statistical significance (Figure 1). Nevertheless, we identified a number of important features. Activation of innate immunity predominated at all stages, except MI-CS stage E.

In these patients, we found a tendency toward a decrease in the frequency of innate immunity activation, both compared with patients with MI-CS stage A (*p* = 0.037, with Bonferroni correction) and patients with MI-CS stage C (*p* = 0.027, with Bonferroni correction). In patients at stage A, neither immune deficiency nor areactive states were observed (Figure 1).

### 3.3. Systemic Inflammation Indices in MI-CS Patients

Calculation of systemic inflammation indices allowed us to identify significant differences between patients at different stages of MI-CS according to the SCAI scale. Significant differences were revealed in PLR (*p* = 0.027), SII (*p* = 0.024), AISI (*p* = 0.010): at stage E we observed their decrease, with the most significant decrease in the AISI index (1058.6 (459.2; 2210.8) at stage C vs. 427.6 (79.9; 773.92) at stage E (*p* < 0.001)) (Figure 2). The PLR and SII indices also decreased: PLR from 117.3 (70.9; 171.4) at stage C to 61.8 (36.5; 101.8) at stage E (*p* = 0.001); SII from 1021.9 (559.0; 1881.2) at stage C to 363.1 (155.4; 1234.9) at stage E (*p* = 0.002) (Figure 2).

Only for patients with MI-CS stage C, we confirmed the presence of correlations between myocardial damage biomarkers and inflammatory indices. The greatest number of correlations were found for the SIRI index (with CK-MB (r = 0.337; *p* = 0.001); with CK (r = 0.334; *p* = 0.001); with troponin I (r = 0.264; *p* = 0.036)). The AISI index correlated with the content of CK (r = 0.256; *p* = 0.018) and CK-MB (r = 0.253; *p* = 0.020). AISI also tended to correlate with the concentration of cardiac-specific troponin I (r = 0.225; *p* = 0.077).

In patients with MI-CS stage E, inverse correlations were found between lactate concentration at the onset of MI-CS and the following inflammation indices: PLR (r = −0.455; *p* = 0.019); SII (r = −0.419; *p* = 0.033); AISI (r = −0.430; *p* = 0.029). The same inflammation indices directly correlated with the time of onset of shock symptoms since admission: PLR (r = 0.505; *p* = 0.010); SII (r = 0.462; *p* = 0.020); AISI (r = 0.412; *p* = 0.041).

### 3.4. Systemic Inflammation Indices May Be Used for Prognosis During MI-CS

In a subgroup of patients with MI-CS stage C who were available for both in-hospital observation and long-term follow-up, we analyzed the relationship between survival rates and systemic inflammation indices. General mortality among the patients with MI-CS was 20.4% after 1 month; 25% after 6 months; 31.8% after 1 year and 43.2% after 2 years of observation. The characteristics of surviving and deceased patients at stage C are presented in Table 4.

We analyzed the relationships between systemic inflammation indices and two-year patient survival (Table 5). Deceased patients exhibited higher values of the MLR index and its enhanced form, SIRI, which incorporated not only monocyte and lymphocyte counts but also neutrophil counts (Table 5).

A more detailed analysis of patient survival in the short and long term demonstrated that lower SIRI values were associated with higher patient survival, both after 1 month (2.6 [1.7; 7.3] in survivors vs. 5.8 [4.6; 6.4] in deceased patients, *p* = 0.037) and after one year of follow-up (2.4 [1.6; 5.2] in survivors vs. 5.5 [2.7; 7.3] in deceased patients, *p* = 0.046).

The SIRI median value in the overall group of stage C patients with follow-up data was 3.34. Dividing patients into subgroups based on this median, patients with SIRI < 3.34 exhibited significantly higher two-year survival compared with those with SIRI ≥ 3.34, as confirmed by the log-rank test (*p* = 0.006) (Figure 3).

### 3.5. Predictors of SIRI Values in MI-CS Patients

In order to study which factors contributed to the increase of SIRI in patients with cardiogenic shock, a multivariate logistic regression analysis was performed. According to the analysis results, the main factors contributing to the increase in SIRI ≥ 3.34 were the heart rate at MI-CS development and the type of immune reaction represented by the predominant activation of innate immunity. A multiple logistic regression model was created (Table 6). The concentration of CK-MB was not a statistically significant factor by itself, but it increased the significance of the whole model.

The sensitivity of the model was 84%, specificity 70%, AUC = 0.818, *p* = 0.047. The ROC curve is presented in Figure 4. The cut-off of the probability of SIRI increase ≥ 3.34 was 0.193.

## 4. Discussion

In our study, we demonstrated that systemic inflammation indices such as PLR, SII, and AISI effectively differentiate the intensity of the inflammatory response at various stages of MI-CS. Additionally, our results indicate that the SIRI index, a comprehensive marker of inflammation and immune activation, holds prognostic value for predicting mortality in patients at stage C of MI-CS.

The important contribution of inflammation to the development of CS has been proven in numerous studies [9,28,29,30,31]. Patients with CS and elevated hsCRP levels had a higher risk of mortality compared with the comparison group [9,30,31,32]. In the study by Soussi S. et al. (2024) [33], a comprehensive assessment of molecular biomarkers allowed identifying subgroups with “adaptive”, “non-inflammatory”, “cardiopathic”, and “inflammatory” phenotypes among all patients with CS. Patients with the “inflammatory phenotype” were characterized by the highest 28-day mortality [33]. An increase in NLR at day 5 after VA-ECMO was associated with a greater risk of in-hospital mortality compared with those cases where NLR decreased [34].

However, data on changes in the level of inflammation severity depending on the SCAI stages were contradictory. It was found that patients at stages A and B of CS and NLR values ≥ 7 were characterized by a higher level of in-hospital mortality than CS patients at stages B and C and NLR values < 7. In other words, patients at stage B with NLR ≥ 7 had a worse prognosis than patients at stage C and NLR < 7 [16]. On the other hand, Jentzer J.C. et al. (2020) demonstrated that the presence of systemic inflammatory response syndrome (SIRS) aggravated the course of shock, but did not allow prediction of 1-year mortality for CS patients at stage E [28]. In turn, Dettling A. et al. (2024) showed that hsCRP levels changed in patients regardless of SCAI stage, while correlating with lactate concentration [9].

This discrepancy between shock severity per SCAI and systemic inflammation levels may stem from the heterogeneity of patient cohorts in these studies, which often include mixed populations where MI patients constitute approximately 50% [35]. According to other data, the incidence of MI does not exceed a third of all patients with CS, and most cases of CS are a consequence of decompensation of chronic heart failure [36]. It should be recognized that the phenotype of patients with MI is unique and not fully understood. Consequently, the mechanism of inflammation development in MI-CS might be different from that in heart failure or sepsis. Therefore, inflammatory biomarker dynamics in MI-CS are expected to follow a specific pattern related to the clinical status of patients. Our study results support this hypothesis.

According to our data, differences in the severity of the inflammatory response at different SCAI stages become obvious when using indices that incorporate not only leukocyte populations but also platelets, namely PLR, AISI, and SII. Moreover, these indices peaked at MI-CS stage C and subsequently decreased at stage E. We did not find prior reports documenting a decrease in inflammation severity alongside MI-CS progression. However, a reduction in immune cell counts in peripheral blood during advancing systemic inflammation in CS has been previously demonstrated. Thus, in a small study, Cuinet J. et al. (2020) [29] showed decreases in the content of lymphocytes and monocytes in CS patients with SOFA > 10. Patients with MI accounted for 50% (12 people) of the entire sample [29]. Our results show that not only did the total leukocyte count decline, but the ratios among leukocyte populations also shifted as MI-CS progressed, while platelet counts similarly decreased, a factor of notable prognostic importance given the well-established adverse impact of thrombocytopenia in CS patients [37,38]. Pathophysiological background of the decrease of inflammatory indices at the advanced SCAI stages may also comprise activation of immunosuppressive T-regulatory cells (Treg) [39,40] and the presence of sepsis complicating MI-CS [4,41]. Further research is warranted to elucidate the mechanisms driving this phenomenon. Nevertheless, the observed inflammation decline at stage E is undoubtedly an unfavorable prognostic indicator, highlighting the impracticality of universally applying anti-inflammatory therapies in this patient subgroup. In-depth phenotyping of lymphocytes is necessary to identify potential target cells in patients at SCAI stage E.

According to our data, the best index for predicting the prognosis among patients with MI-CS stage C (the most common stage of CS among patients admitted to the intensive care unit (ICU)) is the SIRI index, which requires the values of monocytes, neutrophils, and lymphocytes for its calculation. Thus, SIRI offers a holistic insight into the balance between adaptive immune response effectiveness and inflammatory activity.

Despite the simplicity of its calculation, complex systemic inflammation indices are only beginning to demonstrate their clinical utility in monitoring cardiovascular patients. Determining the prognosis is of greatest importance in patients who have developed acute cardiovascular events. It has previously been shown that SIRI can predict the progression of coronary atherosclerosis in patients with ACS who underwent PCI [42] and the onset of major adverse cardiovascular events in both the short-term [43] and long-term [44,45]. According to Ma Y. and Geng X. (2024), SIRI values were higher in patients with ACS over 60 years of age than in patients under 60 years of age, and it was in this age group that they were associated with an increased risk of MACE [16]. An association of SIRI with GRACE score has also been shown in patients with ACS [46].

We are the first to demonstrate the possibility of using SIRI as a predictor of two-year survival in patients with MI-CS. Moreover, its values were interrelated with the infarction size (according to the data of correlation analysis and logistic regression) and the simplest indicator of compromised hemodynamics—heart rate (according to the data of logistic regression). Also, as expected, the greatest increase in SIRI value was observed when activation of innate immunity prevailed as a leading type of immune reaction. Interconnection between hemodynamics, autonomic nervous system activity, and inflammation are being widely studied [47,48,49,50]. However, the extent to which evaluation of heart rate variability can be applied to identify inflammation in MI-CS patients remains unclear and requires further research.

In general, given that inflammation indices increase with excessive activation of the innate immune system associated with suppression of its adaptive component, it is plausible that follow-up of MI-CS patients at certain stages may benefit not only from general anti-inflammatory therapy as an inflammation correction, but also from the targeted interventions aimed at maintaining immune homeostasis. Moreover, our data support a personalized approach in each specific case. It is necessary to determine at which time points within the in-hospital and remote stages these approaches would be most appropriate.

A number of therapeutic approaches are currently being developed to control excessive inflammatory response. They include both systemic suppression of inflammation and targeting the specific inflammatory mediators [12,35]. The potential therapeutic targets currently being considered for MI-CS patients include but are not limited to IL-6 [51], complement components [52], dipeptidyl peptidase-3 (DPP-3) [53], and C-reactive protein [54].

It is possible to develop therapeutic approaches aimed at restoring immune homeostasis if immune suppression is detected. Immunomodulatory cytokines (IFN-γ, IL-7, -15, -33), growth factors (GM-CSF), thymosin-α, immune checkpoint inhibitors (PD-1/PD-1L, CTLA-4, LAG-3, TIM-1), administration of immunoglobulins, and cell therapy with mesenchymal cells are being studied for use in patients with sepsis [55,56]. The extent to which these approaches are applicable to patients at stage E of MI-CS remains to be elucidated.

The question of the leading source of the development of systemic inflammation in MI-CS remains open. Undoubtedly, a key factor is the excessive release of inflammatory mediators during cardiac tissue damage, which is superimposed on pre-existing chronic subclinical inflammation associated with atherosclerosis. In the case of MI-CS following PCI, an additional myocardial damage occurs after restoration of the blood flow as a result of ischemia/reperfusion injury [57]. Further on, hypoperfusion of internal organs becomes a powerful inflammatory factor in the course of MI-CS development and progression [58]. Thus, when a patient with MI-CS is admitted to the intensive care unit (ICU), the attending physician is dealing with the result of the interaction of all the above-mentioned factors. Moreover, excessive inflammation exacerbates myocardial injury and multiorgan dysfunction. Montero S. and Bayes-Genis A. (2022) proposed a controversial hypothesis suggesting that inflammation may drive multiorgan dysfunction rather than vice versa, coining the term Systemic Inflammation Driven Syndrome (SIDS) as potentially more accurate than SIRS [58]. Perhaps, identifying the critical stage related to the development of a systemic inflammatory response in patients with MI-CS will help to break the vicious circle and to determine the time points at which interventions aimed at interposing the excessive inflammation will be most efficient.

On the other hand, there is no information on whether the strength of the systemic inflammatory response increases together with the severity of MI-CS, nor whether the reverse dynamics of MI-CS are associated with a decrease in the severity of systemic inflammation, for example, when the patient transits from SCAI stage C to SCAI stage B, and possibly further to stage A. This remains unclear and warrants further investigation.

The limitations of this study include its retrospective, single-center design, and the relatively small number of included patients, especially those with two-year survival data. As is common in studies on MI-CS in clinical settings, an additional limitation is that systemic inflammation indices were assessed only upon admission to the intensive care unit. Moreover, upon admission, different patients were obviously at different stages of the pathological condition (the minimum “pain-to-door” time in the patients’ sample was 1 min; the maximum was 72 h). In addition, the present study did not include an assessment of high-sensitivity C-reactive protein (hsCRP) concentration, a well-established marker of systemic inflammation in ACS [59,60]. This was due to the fact that at the time of writing the article, the registry included patients admitted to the clinic in 2020, when hsCRP assessment was not a mandatory part of the protocol of the general clinical study. The prospects for diagnosing systemic inflammation in MI-CS are only beginning to unfold, highlighting the need for unified laboratory scales and a set of mandatory laboratory markers. In our opinion, systemic inflammation indices, especially those proven significant in prognostic assessment of MI-CS patients, should be integrated alongside hsCRP into such future diagnostic scales. Another study limitation is that there was no possibility of excluding patients with acute infection at the time of MI-CS diagnosis. The contemporary follow-up protocols did not consider this factor; however, this might have compromised the degree of inflammation and could have affected the values of the indices under evaluation. Further studies, including both retrospective and prospective blocks, will allow more complete characterization of the molecular–cellular “portrait” of patients with MI-CS in relation to the development of immune response and systemic inflammation, as well as the development of personalized approaches to diagnosis and treatment in this difficult cohort of patients.

Our study reinforces that a one-size-fits-all approach simply does not cut it in patients with MI-CS when targeting inflammation. Patients with MI-CS exhibit a spectrum of differences across various SCAI stages, necessitating a personalized approach to therapeutic care. Evaluation of the complete blood count at the onset of MI-CS, followed by calculation of PLR, SII, AISI, and SIRI, not only allows stratification of patients according to their inflammatory phenotype but also provides valuable prognostic insight.

## 5. Conclusions

According to the results of our study, there is a distinct pattern in the dynamics of systemic inflammation indices in patients with MI-CS depending on the SCAI scale: patients at stage C are characterized by the highest values of the PLR, AISI, and SII indices, while patients at stage E show a decrease in these values, indicating suppression of systemic inflammation and a risk of immune paralysis development. The SIRI index has the greatest prognostic significance in the mid-term for patients at stage C of MI-CS, with its value dependent on both infarction size and hemodynamic characteristics, as well as the type of immune response in individual patients. Further studies are needed to confirm the significance of the proposed inflammatory indices and to identify specific cellular or molecular targets for restoring immune homeostasis during MI-CS, which could assist intensive care teams in diagnostic and therapeutic decisions when managing this complex patient group.

## Figures and Tables

**Figure 1 jcm-14-04283-f001:**
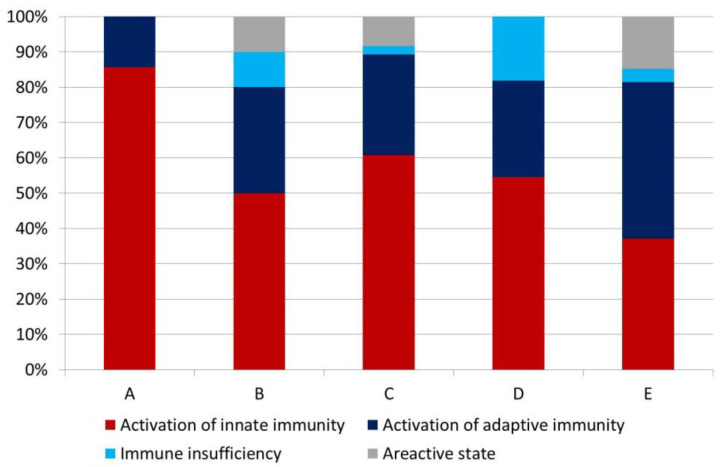
Types of immune reactions in patients with myocardial infarction complicated by cardiogenic shock.

**Figure 2 jcm-14-04283-f002:**
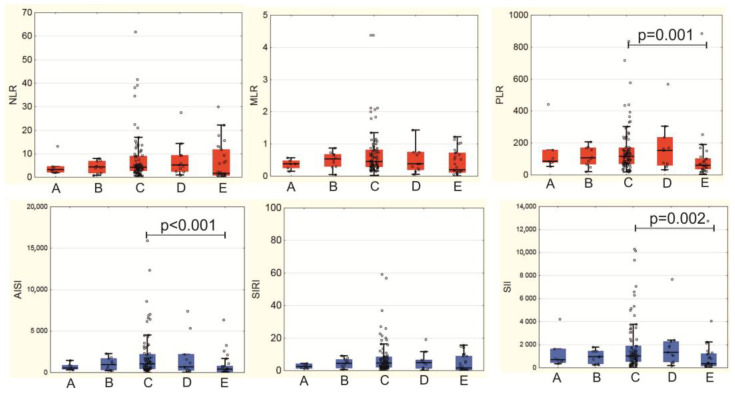
Systemic inflammation indices depending on the stages of myocardial infarction complicated by cardiogenic shock; simple ratios are marked with red color; complex indices are marked with blue color.

**Figure 3 jcm-14-04283-f003:**
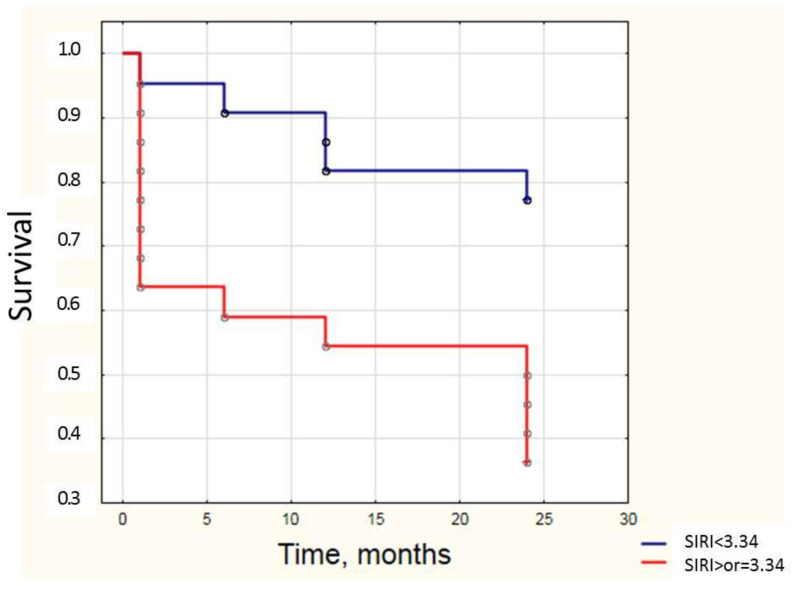
Kaplan–Meier curves for survival of myocardial infarction complicated by cardiogenic shock patients, SCAI stage C, with various levels of SIRI.

**Figure 4 jcm-14-04283-f004:**
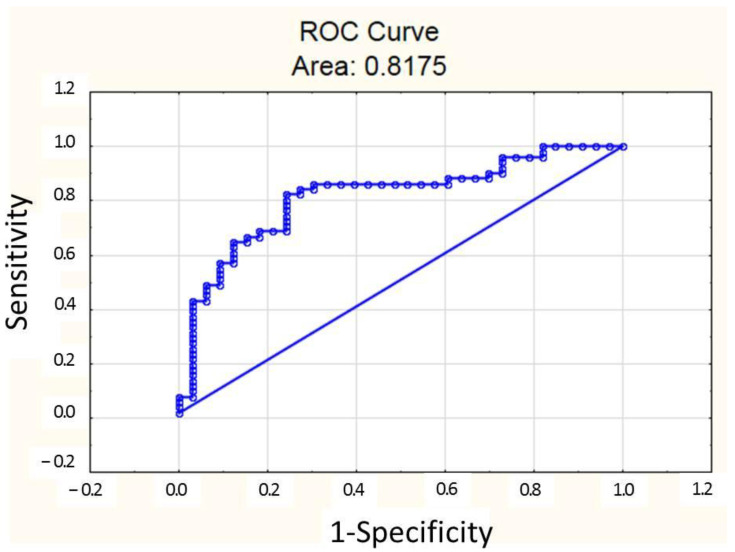
ROC curve of multiple logistic regression for classification of patients into groups with SIRI < 3.34 and SIRI ≥ 3.34. Upper line is based on the values of the model probability. Lower line represents reference line.

**Table 1 jcm-14-04283-t001:** Definition of the types of immune reactions.

Parameters		Lymphocytes, Absolute Counts		
		Below the Reference Range	Within the Reference Range	Above the Reference Range
**Leucocytes, absolute counts**	**Above the reference range**	Innate immunity activation	Innate immunity activation	Adaptive immunity activation
	**Within the reference range**	Immune insufficiency	Areactive state	Adaptive immunity activation
	**Below the reference range**	Immune insufficiency	Immune insufficiency	Adaptive immunity activation

**Table 2 jcm-14-04283-t002:** Clinical and anamnestic characteristics of patients with myocardial infarction complicated by cardiogenic shock.

Parameter		SCAI Stage A (*n* = 7)	SCAI Stage B (*n* = 10)	SCAI Stage C (*n* = 84)	SCAI Stage D (*n* = 11)	SCAI Stage E (*n* = 27)	*p*
Age, years		68 (58; 71)	66 (59; 74)	73 (65; 81)	80 (69; 82)	80 (75; 87)	0.004
Men, %		71.4	100.0	47.6	54.5	33.3	0.006
Arterial hypertension, %		85.7	100.0	94.0	90.9	92.6	0.811
Diabetes mellitus, %		42.9	10.0	32.1	45.5	14.8	0.195
Obesity, %		42.9	40.0	28.6	18.2	85.2	0.655
STEMI/NSTEMI, %		71.4/28.6	90.0/10.0	77.4/22.6	81.8/18.2	70.4/29.6	0.757
“Pain-to-door” time, minutes		-	149 (100; 374)	220 (137; 540)	324 (139; 1080)	153 (90; 410)	0.134
“Door-to-balloon” time, minutes		-	84 (50; 120)	63 (52; 145)	114 (44; 230)	51 (43; 115)	0.130
PCI, *n* (%)		5 (71.4)	6 (60.0)	55 (65.5)	9 (81.8)	14 (51.9)	0.710
TIMI 3 flow in PCI, *n* (%)	restored	3 (42.8)	7 (70)	38 (45.2)	8 (72.7)	9 (33.3)	0.999
	not restored	1 (14.4)	1 (10)	16 (19.1)	2 (18.2)	5 (18.5)	
	no data	3 (42.8)	2 (20)	30 (35.7)	1 (9.1)	13 (48.2)	
Maximum troponin I level, ng/mL		0.47 (0.15; 1.30)	0.49 (0.04; 6.4)	0.50 (0.06; 3.20)	5.90 (0.61; 14.00)	0.37 (0.09; 6.32)	0.336
Vasocative inotropic score, points		-	7.0 (5.0; 10.0)	7.5 (5.0; 26.0)	30.0 (8.0; 50.0)	40.0 (5.0; 62.0)	0.999
Lactate, mmol/L		1.9 (1.8; 1.9)	1.7 (1.3; 1.7)	3.0 (2.4; 5.1)	4.6 (2.6; 6.0)	8.7 (6.9; 11.4)	<0.001
pH (venous)		7.41 (7.21; 7.44)	7.39 (7.33; 7.42)	7.32 (7.27; 7.35)	7.30 (7.26; 7.31)	7.14 (7.06; 7.19)	<0.001
Hb, g/L		144 (113; 151)	148 (119; 156)	134 (116; 145)	128 (116; 137)	120 (103; 138)	0.125
Central venous pressure, cm H_2_O		5.0 (4.0; 6.0)	9.0 (6.0; 10.0)	12.0 (8.0; 15.0)	11.5 (9.0; 13.0)	16.0 (8.0; 19.0)	0.062
Number of arteries with stenosis, %:							
1 artery		14.3	18.5	20.2	18.2	18.5	0.958
2 arteries		42.9	14.8	27.4	36.4	14.8	
3 arteries		42.9	33.3	40.5	45.5	33.3	
Mortality, %		71.4	40.0	44.0	63.6	85.2	0.003

Hb, hemoglobin; NSTEMI, non-ST-elevation myocardial infarction; PCI, percutaneous coronary intervention; STEMI, ST-elevation myocardial infarction; TIMI, thrombolysis in myocardial infarction; *p*-level calculated according to Kruskal–Wallis ANOVA test.

**Table 3 jcm-14-04283-t003:** Complete blood count test data in patients with myocardial infarction complicated by cardiogenic shock.

Parameters	SCAI Stage A (*n* = 7)	SCAI Stage B (*n* = 10)	SCAI Stage C (*n* = 84)	SCAI Stage D (*n* = 11)	SCAI Stage E (*n* = 27)	*p*
Leucocytes, ×10^9^/L	10.5 (10.4; 12.8)	11.6 (9.4; 13.7)	13.7 (10.3; 15.9)	11.8 (9.9; 17.1)	14.1 (8.4; 17.8)	0.452
Erythrocytes, ×10^12^/L	4.5 (3.9; 5.1)	4.5 (4.2; 5.2)	4.4 (4.0; 5.0)	4.5 (4.1; 4.8)	4.3 (3.6; 5.0)	0.697
Platelets, ×10^9^/L	217 (186; 319)	221 (188; 249)	244 (175; 297)	208 (186; 366)	188 (139; 247)	0.108
Neutrophils, ×10^9^/L	7.7 (6.5; 9.5)	8.5 (6.9; 10.6)	9.7 (7.1; 13.0)	8.0 (6.7; 14.8)	6.8 (3.6; 11.7)	0.104
Lymphocytes, ×10^9^/L	2.5 (1.8; 2.6)	1.9 (1.5; 3.5)	2.1 (1.5; 3.1)	1.8 (1.1; 3.2)	2.7 (1.1; 6.6)	0.635
Monocytes, ×10^9^/L	0.98 (0.68; 1.21)	0.99 (0.72; 1.28)	1.02 (0.67; 1.31)	0.55 (0.44; 0.89)	0.81 (0.49; 0.99)	0.022

**Table 4 jcm-14-04283-t004:** Clinical and anamnestic characteristics of patients with myocardial infarction complicated by cardiogenic shock (SCAI stage C), according to survival rate.

Parameter	SCAI Stage C, Alive Patients (*n* = 25)	SCAI Stage C, Deceased Patients (*n* = 19)	*p*
Age, years	70 (61; 75)	75 (68; 81)	0.028
Men (%)	14 (56.0)	11 (57.9)	0.999
Arterial hypertension (%)	25 (100)	18 (94.7)	0.432
Diabetes mellitus (%)	8 (32)	8 (42.1)	0.999
Obesity (%)	10 (40)	7 (36.8)	0.999
STEMI/NSTEMI (%)	21/4 (84.0/16.0)	12/7 (63.2/36.8)	0.164
“Pain-to-door” time, minutes	185 (87; 335)	288 (165; 720)	0.323
“Door-to-balloon” time, minutes	68 (54; 118)	59 (52; 102)	0.538
Vasocative inotropic score, points	0.07 (0.04; 0.22)	0.67 (0.05; 1.50)	0.309
Lactate, mmol/L	7 (3; 10)	6 (2; 10)	0.917
pH (venous)	3.0 (2.4; 5.8)	3.0 (2.4; 5.0)	0.689
Hb, g/L	7.31 (7.27; 7.34)	7.32 (7.23; 7.33)	0.657
Central venous pressure, cm H_2_O	140 (129; 153)	125 (100; 138)	0.025
Vasocative inotropic score, points	10.0 (8.0; 15.0)	11.5 (7.5; 15.0)	0.750

**Table 5 jcm-14-04283-t005:** Complete blood count data and systemic inflammation indices in patients with myocardial infarction complicated by cardiogenic shock (SCAI stage C) stratified by survival rate.

Parameter	SCAI Stage C, Alive Patients (*n* = 25)	SCAI Stage C, Deceased Patients (*n* = 19)	*p*
Leucocytes, ×10^9^/L	12.3 (9.6; 15.8)	13.4 (10.3; 15.1)	0.870
Erythrocytes, ×10^12^/L	4.7 (4.1; 5.0)	4.2 (3.9; 4.6)	0.103
Platelets, ×10^9^/L	243 (201; 281)	288 (175; 358)	0.511
Neutrophils, ×10^9^/L	9.4 (6.1; 11.5)	9.5 (7.4; 16.3)	0.279
Lymphocytes, ×10^9^/L	2.6 (1.9; 3.2)	2.2 (1.9; 2.9)	0.212
Monocytes, ×10^9^/L	0.91 (0.64; 1.20)	1.02 (0.68; 1.42)	0.411
NLR	3.20 (2.33; 4.92)	3.96 (3.44; 9.22)	0.119
MLR	0.29 (0.24; 0.45)	0.47 (0.34; 0.70)	0.046
PLR	97.4 (71.4; 128.2)	136.5 (74.9; 225.2)	0.157
AISI	600 (363; 1368)	1430 (546; 2934)	0.093
SIRI	2.38 (1.61; 5.23)	5.47 (2.69; 7.28)	0.036
SII	853 (541; 1298)	1495 (440; 3445.9)	0.089

**Table 6 jcm-14-04283-t006:** Logistic regression model coefficients and their significance levels.

Parameters	Estimates	*p*
Intercept	−2.74	0.008
Heartbeat	0.02	0.020
CK-MB	0.01	0.142
Activation of innate immunity	1.49	0.007

## Data Availability

The data used to obtain results presented in the study are available on reasonable request.

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
