# Peer review of "Inflammatory Indices in Patients with Myocardial Infarction Complicated by Cardiogenic Shock, and Their Interconnections with SCAI Stages and Patients’ Survival: A Retrospective Study"

_jcm, 2025, doi:10.3390/jcm14124283_

Round 1

Reviewer 1 Report

Comments and Suggestions for Authors

To the Authors,

Thank you for submitting your manuscript on systemic inflammatory indices in cardiogenic shock after myocardial infarction. This is an interesting topic which, although well known, still deserves further investigation as it may open new therapeutic avenues.

Overall, the manuscript is well executed. You have previously published other aspects of this cohort, which demonstrates your deep familiarity with these patients.

  • Introduction: In my opinion, the introduction is too long. While it may help readers unfamiliar with the topic, it might be excessive for a journal article; however, I leave this to your and the editors’ discretion.
  • Discussion: Similarly, the discussion includes some topics that fall outside the immediate scope of this study. Nevertheless, the information is sound, and its length can be decided by you and the editorial team based on journal limits.
  • Results and Figures: The presentation of results and figures is clear and easy to understand.
  • Conclusions: Appropriate and well supported by the data.
  • The bibliography is extensive, up to date, and relevant to the subject matter.

Minor issues:

  • Table 4: One parameter remains in Russian.
  • References: The DOIs for references 13, 15, 17, and 19 each contain an extra final digit, rendering them incorrect.

I have nothing further to add, and I congratulate you on this work.

Kind regards,

Author Response

Dear reviewer!

We sincerely thank you for your time and efforts that you spent working on our manuscript. We are grateful for the high evaluation of our work and we have tried our best to address the concerns that have been raised. Please, find the elaborated answers to your comments below. We have marked the introduced changes with yellow color in the manuscript.

  1. In my opinion, the introduction is too long. While it may help readers unfamiliar with the topic, it might be excessive for a journal article; however, I leave this to your and the editors’ discretion.

We have shortened parts of introduction, devoted to the pathogenesis of myocardial infarction and description of inflammatory indices.

  1. Similarly, the discussion includes some topics that fall outside the immediate scope of this study. Nevertheless, the information is sound, and its length can be decided by you and the editorial team based on journal limits.

We tried to omit elaborate description of the topics that are not related to the immediate scope of the manuscript, such as the possible background of the revealed changes in the values of inflammatory indices and associations between inflammation and hemodynamics.

  1. Table 4:One parameter remains in Russian

We apologize for this inconvenience. We have corrected Table 4

  1. The DOIs for references 13, 15, 17, and 19 each contain an extra final digit, rendering them incorrect.

We removed the final digits from the references

  1. Also, we referred for the help of the English-speaking colleague for editing of the language of the manuscript. We hope that editorial changes improved the readability and transparency of the manuscript, and made it more explicit.

Reviewer 2 Report

Comments and Suggestions for Authors General Comments:   This paper focuses on the immune responses following myocardial infarction, with particular attention to their variation across different clinical severities. The authors included a heterogeneous cohort of patients (i.e., STEMI and NSTEMI) and stratified immune responses according to the SCAI shock classification. They found that in patients at more advanced SCAI stages, the innate immune response is more dominant, whereas in the most severe stages, features of immune suppression and anergy become more prevalent. Conversely, patients in earlier stages show a more prominent adaptive immune response and a reduced innate reaction. I would like to congratulate the authors on this well-written and generally well-conducted study.   Minor Comments:
  • Please revise Table 4, particularly the labeling, for improved clarity and consistency.
  • Did you exclude patients with active infections at the time of MI diagnosis? This may influence immune profiling.
  • Consider condensing the discussion section to enhance readability and focus on the key findings.

Author Response

Dear reviewer!

We sincerely thank you for your time and efforts that you spent working on our manuscript. We are grateful for the high evaluation of our work and we have tried our best to address the concerns that have been raised. Please, find the elaborated answers to your comments below. We have marked the introduced changes with yellow color in the manuscript.

  1. Please revise Table 4, particularly the labeling, for improved clarity and consistency.

We apologize for the inconvenience with the Table labeling. We have corrected Table 4

  1. Did you exclude patients with active infections at the time of MI diagnosis? This may influence immune profiling.

Active infections were indeed not excluded at the time of MI diagnosis. We have added this among the limitations of the study:

“Another study limitation is that there was no possibility to exclude patients with acute infection at the time of MI-CS diagnosis. The contemporary follow-up protocols do not consider this factor; however it might compromise the degree of inflammation and could have affected the values of indices being under evaluation.” (Lines 447-451).

  1. Consider condensing the discussion section to enhance readability and focus on the key findings.

We tried to omit elaborate description of the topics that are not related to the immediate scope of the manuscript, such as the possible background of the revealed changes in the values of inflammatory indices and associations between inflammation and hemodynamics.

  1. Also, we referred for the help of the English-speaking colleague for editing of the language of the manuscript. We hope that editorial changes improved the readability and transparency of the manuscript, and made it more explicit.
